



# Check dam impact on sediment loads: example of the Guerbe River in the Swiss Alps - a catchment scale experiment

**Ariel Henrique do Prado**[1], David Mair[1], Philippos Garefalakis[1], Chantal Schmidt[1],

Alexander C. Whittaker[2], Sebastien Castelltort[3] and Fritz Schlunegger[1]

[1] University of Bern, Institute of Geological Sciences, Bern, Switzerland

[2] Imperial College, Department of Earth Science and Engineering, London, United Kingdom

[3] University of Geneva, Department of Earth Sciences, Geneva, Switzerland

*Correspondence to*: Ariel do Prado (ariel.doprado@geo.unibe.ch)

**Abstract**

The construction of check dams is a common human practice around the world where the aim is to reduce the damage by flooding events through mountain streams. However, quantifying the effectiveness of such engineering structures has remained very challenging and requires well-selected case studies, since the outcome of such an evaluation depends on site specific geometric, geologic, and climatic conditions. Conventionally, the check dams' effectiveness has been estimated using information about how the bedload sediment flux in the stream changes after the check dams are constructed. A permanent lowering of the bedload flux not only points to a success in reducing the probability of sediment transport occurrence but also implies that the sediment input through the system is likely to decrease. Here, we applied two methods (Meyer-Peter Müller versus Recking approach) to estimate and compare the sediment transport in a mountain stream in Switzerland under engineered and non-engineered conditions. Whereas the first approach is a classical equation that is based on flume experiment data with a slope less than 0.02 m/m, the second approach (Recking) has been deviated based on bedload data acquired from active mountain streams under steeper conditions. We selected the Guerbe River situated in the Swiss Alps as a case study, which has been engineered since the end of the 19[th] century. This has resulted in more than 110 check dams along a c. 5 km reach where sediment has continuously been supplied from adjacent hillslopes, primarily by landsliding. We measured the riverbed grain size, topographic gradients, and river widths within selected segments along this reach. Additionally, a gauging station downstream of the check dams yielded information to calibrate the hydroclimatic situation for the study reach, thus yielding ideal conditions for our catchment-scale experiment. Using the acquired data and the historical runoff dataset covering the time interval between 2009 and 2021 and considering the current engineered conditions, we estimated a mean annual volume of transported bedload which ranges from 900 to 6'000 m³ yr⁻¹. We then envisaged possible channel geometries before the check dams were constructed. We inferred (1) higher energy gradients which we averaged over the length of several check dams and which we considered as a proxy for the steeper river slope under natural conditions; (2) channel widths that are smaller than those measured today, thereby anticipating that the channel was more confined in the past due to the lateral supply of sediments through landsliding; and (3) larger grain size percentiles, which we consider to be similar to the values measured





from preserved landslides in the region. Using such potential non-engineered scenarios as constraints, the two

equations both point towards a larger sediment flux compared to the engineered state, although the results of these equations differed significantly in magnitude. Whereas the Recking approach returned estimates where the
bedload sediment flux is c. 10 times larger in comparison with the current situation, the use of the Meyer-Peter Müller equation predicts an increase of c. 100 times in bedload fluxes for a state without check dams. These
results suggest that the check dams in the Guerbe (Gürbe) River are highly efficient not only in regulating sediment transport by decreasing the probability of high sediment flux occurrence during torrential conditions,
but also in stabilizing the channel bed by avoiding incision. The most likely consequence is a stabilization of the terrain around such structures by reducing the landslide occurrence.
**1. Introduction**

Engineering structures known as check dams have been constructed in many mountainous streams

around the world with the intention to mitigate hazards caused by the transfer of large volumes of sediment in relation to flooding, landsliding and debris flows (Piton et al., 2017; Lucas-Borja et al., 2021). Check dams are
transversal structures built across the channel bed and made of wood, rock or concrete. They create space that can initially store sediment derived from farther upstream. Subsequently, this space is filled with material,
which diminishes its capacity to store additional sedimentary material. However, even in their filled stages, the check dams seem to remain operational for two reasons. First, they prevent the stream from further incising into
substratum, which in turn contributes to the stabilization of landslides and the preservation of soils on the bordering hillslopes; and second, they reduce the stream's capacity to evacuate the supplied sedimentary
material due to a reduction of the channel's friction slope (Castillo et al., 2014; Piton et al., 2017). Although it is generally appreciated that the construction of check dams is beneficial for reducing risks, it has been a recurring
challenge for engineers and the different stakeholders to take decisions about whether or not to install such infrastructure because of the high maintenance costs (e.g., Jackle, 2013; Ramirez, 2022) and also because of bio-
environmental concerns (Bombino et al., 2014). Furthermore, in most of these streams, the construction of check dams started before a survey on sediment flux was conducted, with the consequence that information
about the pre-engineered conditions on sediment discharge is not available (Piton et al., 2017). Hence, it remains difficult to quantify the efficiency of such infrastructure, and society is left with limited information for taking
decisions on whether or not to build new check dams and/or to maintain older ones. Under these circumstances, an indirect method of estimating the contribution of check dams to reduce risks is needed for stakeholders when
they have to take evidence-based decisions on how to manage such infrastructure. In the past decade, Castillo et al. (2014) developed a model to estimate the efficiency of check dams. They focussed on exploring how the
variations of the friction slope angles, which varied through changing the spacing between the dams, impacted the flow regime. However, since the friction slope is not the only variable that controls the transport of sediment
(e.g., Meyer-Peter and Müller, 1948; Piton and Recking, 2016; Recking et al., 2016; Wong and Parker, 2006), data on slope changes alone is not sufficient to fully appreciate and predict possible reductions of risks when
check dams are set in place. As an alternative approach, estimates of the sediment volumes transported on the riverbed could be used to predict the efficiency of check dams once the space behind them has been filled



(Kaitna et al., 2011; Piton et al., 2017; Keiler and Fuchs, 2018). Therefore, available bedload equations that were calibrated on data acquired in active streams and flume experiments (e.g. Meyer-Peter and Müller, 1948;
Piton and Recking, 2016; Recking et al., 2016) are potential tools for such an evaluation, and their application depends on variables that can be measured in the field (e.g., slope, width, and grain size distributions).
To do so, we studied the Guerbe River, which is a torrent situated on the northern margin of the Swiss Alps (Fig. 1). There, the c. 5 km-long headwater reach has experienced a >100-year-long history of check dam
construction and maintenance. The first ones were installed during the 19th century and mainly consisted of structures made of wood and stone (Salvisberg, 2017). Subsequently, they were replaced by reinforced concrete
dams in the 20th century, forming steps that are up to 10 m high (e.g., Fig. 1b). However, during several events along their history, the check dams failed and released a large amount of material to downstream of the channel
generating a large loss to the local society (Salvisberg, 2017). After the last failure event, which occurred in January 2018 with the displacement of the c. $4.5 \times 10^6$ m³-large Meierisli landslide that damaged >10 of these
check dams (Andres and Badoux, 2019), the local community has been confronted with taking a decision on how to manage this situation in the future without a-priori, physics-based information on the efficiency of this
infrastructure. Therefore, this paper aims to offer such a quantitative evaluation. Here, we estimate the efficiency regarding the transport of bedload material for a staircase of check dams using the Guerbe River as a
natural laboratory. We collect high-resolution data on the channel's metrics (slope, width) and the grain size distribution in the field, and we combine this data with information about the hydroclimatic properties of the
Guerbe River basin. The scope is to estimate the modern bedload sediment flux for the current engineered state. These results are then compared with the outcome of model runs where pre-engineered conditions regarding
channel metrics (slope, width) and grain size distributions are considered.

**2. Local setting**

The studied reach of the Guerbe River (Fig. 1a), which is situated at the northern border of the Swiss Alps, can be segmented into four parts: (1) The headwater reach, which is the uppermost segment covering an
area of c. 5 km², is characterized by a dendritic network made up of first to third-order channels. The stream originates in the Gantrisch area at an altitude of c. 1800 m a.s.l. where the bedrock is made up of steeply dipping
limestones, dolostones and marls that are part of the Penninic Klippen belt (Jäckle, 2013). Towards the lower part of the headwater reach, the Mesozoic units are covered by several meters-thick glacial till. This headwater
reach transitions into a steep segment at an elevation of c. 1200 m a.s.l. where the longitudinal stream profile of the Guerbe River shows a knickpoint (next to site 1 in Fig. 1 and circle in Fig. 2). The occurrence of such a
knickpoint in the stream profile is also seen in the morphology of the bordering hillslopes where slope angles are c. 20-25° steep. These hillslopes constitute an important sediment source of the Guerbe River. Uphill, these
hillslopes mark a sharp transition towards a flatter landscape that was originally formed by glaciers, thereby defining also a knickzone on the hillslopes (Fig. 2). The second segment occurs downstream of this knickzone
area, where the Guerbe River has been fully engineered by > 60 check dams. There, the bedrock comprises a suite of Late Cretaceous to Paleocene Gurnigel Flysch and the Early Oligocene Lower Marine Molasse
(L.M.M.) units, both of which are alternations of shales and sandstones. They are dissected by multiple





landslides along the entire c. 2 km-long second segment of the Guerbe River (Red segment in Fig. 1). These

landslides either originate >1 km upstream of the Guerbe channel and are deep-seated with a decollement

horizon up to 20 m below the surface (Thuner Tagblatt, 25th of Mai 2018), or they border the Guerbe trunk

stream as a few shallow-seated and < 100 m-long features (decollement < 2 m deep) as own observations have

shown. Along this second reach, the Guerbe River shows a "colluvial" stream pattern as defined by Piton and

Recking (2017). The third segment comprises the reach along which the river then transitions on a c. 4 km²-

large alluvial fan where the apex is located at an elevation of c. 800 m a.s.l (white segment in Fig. 1). The

stream remains channelized and with presence of check dams on the entire fan. In the final segment, the stream

enters the floodplain area, where it flows in a confined channel until its confluence with the Aare River c. 20 km

farther downstream.

The climate in the region is typical for a pre-alpine region with a mean annual precipitation rate that

ranges between 2000 mm yr⁻¹ in the mountains and 1100 mm yr⁻¹ at lower elevations (Ramirez et al., 2022).

Accordingly, the mean annual water discharge is c. 1.3 m³ s⁻¹ as recorded by the Burgistein gauging station c. 4

km downstream of the source area, and the maximum discharge during the past 22 years has been 84 m³ s⁻¹,

measured on the 29th of July in 1990 (Ramirez et. al, 2022). Peak water flux occurs either during convective

thunderstorms in summer or during periods of extended precipitation in late spring and fall. In addition, a

denudation rate of c. 260 mm/kyr on our surveyed catchment was estimated from ¹⁰Be concentrations obtained

in the Guerbe River (Delunel et al., 2020).

### 3. Methods and datasets

**3.1. Flow specificities related to check dams**

One important functioning of the filled check dams is to reduce the kinetic energy of a mountain

stream, which in turn is expected to reduce the sediment load (Castillo et al., 2014). In the reach downstream of

a check dam, the largest energy dissipation occurs when the water that falls from the check dam spillway

impacts the ground. The water enters a high-turbulent flow stage, thereby creating a scour and thus a pool just at

the foot of the check dam (e.g. Fig. 3). A second contribution to the energy dissipation derives from the basal

friction exerted by the arrangement of clasts along the river bed as the water leaves the pool (Piton and Recking,

2016). The flow is then more uniform, and local turbulences occur less frequently. The spacing between two

adjacent check dams can affect this pattern when the distance is shorter than $30h_c$ (where $h_c$ is the critical depth

for which the Froude number is equal to 1; Piton and Recking, 2016), which is not the case for the Guerbe River

since the spacing between the check dams is > 20 m and the maximum critical flow depth is 0.43 m at the apex

of the alluvial fan (calculation done by using the measurements presented in the results section). This

assumption is key for the application of the bedload equations presented in section 3.2 since it requires the

occurrence of a uniform flow.





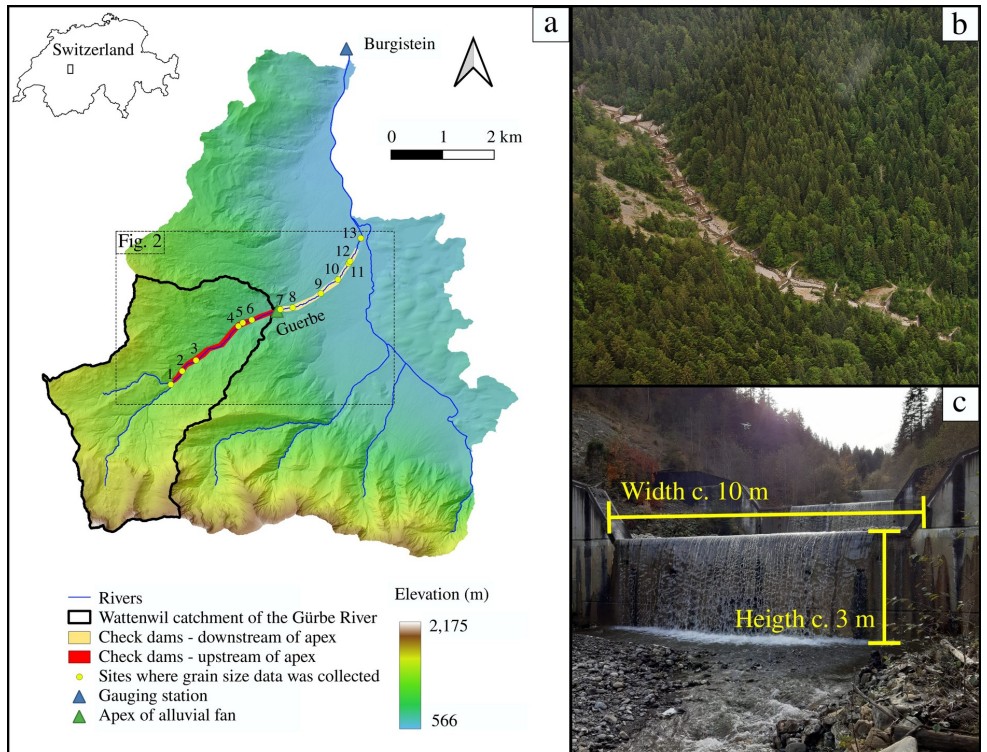

**Figure 1.** (a) DEM of the Guerbe catchment upstream of the Burgistein gauging station, and sub-catchment where sediment has been produced and supplied to the trunk channel (Wattenwil catchment of the Guerbe River). The dashed rectangle limits the area shown in Fig. 2. (b) Aerial picture in the Guerbe River with the staircase check dams. Additionally, the picture shows a steep non-vegetated area where recent hillslope instabilities have prevented a dense vegetation cover to establish (c) Example of check dams with heights of c. 3 m.

### 3.2. Bedload discharge in mountain streams

Available empirical equations to estimate the volumes of bedload transported by streams have generally been developed for rivers with slopes < 0.02 m m$^{-1}$ (or 1.2 degrees) and riverbed material composed of grains with sizes that range from coarse sand to coarse gravel (e.g. Bagnold, 1980; Einstein, 1950; Meyer-Peter and Müller, 1948; Parker, 2008; Recking et al., 2012; Wong and Parker, 2006). However, mountain streams usually have steep slopes (> 0.02 m m$^{-1}$) and transport material with large sizes ranging from gravel to boulders (Piton and Recking, 2016; Recking et al., 2016). In addition, the transport of sediment is controlled by flow-specific conditions that limit a direct application of a large number of the available equations to such a case (Rickenmann, 2001). One option for estimating the bedload flux is to select one of the available equations that are based on the Meyer-Peter and Müller (1948) formula, here referred to as MP.M., and to consider a





correction for the critical Shields shear stress along river reaches where the slopes are steeper than 0.02 m m⁻¹ (Lamb et al., 2008; Recking et al., 2012; Shvidchenko et al., 2001). An alternative is the formulation proposed
by Recking (2013), recently reformulated by Recking et al. (2016). This formula considers different channel morphologies and was evaluated and validated for steep and coarse-grained mountain streams by Piton and
Recking (2017). Therefore, this equation may be better suited for estimating the bedload flux in the Guerbe River. We thus applied the equations developed by Recking et al. (2016) to calculate the bedload flux in the
stream for both the engineered and non-engineered conditions. We additionally conducted the same calculations but employed the Wong and Parker (2006) formulation instead, which is an updated and corrected version of the
Meyer-Peter and Müller (1948) formula. We justify the selection of the MP.M. approach since it is one of the most frequently used equations in the literature to compute the transport of bedload. Therefore, the results from
this equation will be used here as a benchmark when we compare both sets of formulas. Yet we note that we consider the Recking equation as the best formula to calculate the sediment bedload in steep mountain streams.

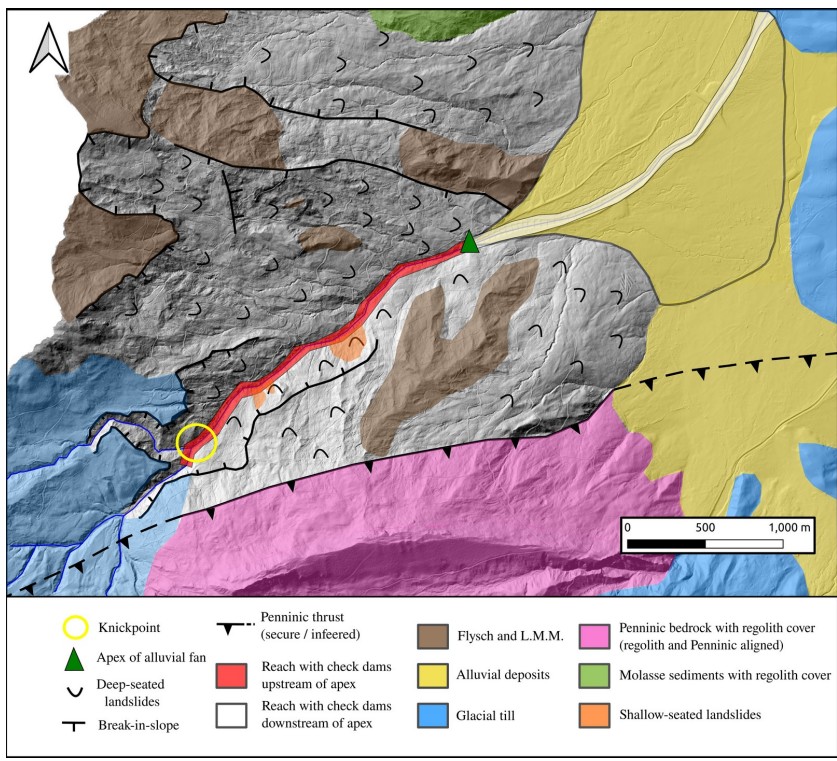


**Figure 2.** Map of the landslides and incised areas of the Guerbe River together with the geology underlying the

catchment. The position of the knickpoint is marked inside the yellow circle. See Fig. 1 for the position of this map. Here the LMM is referred to as the Lower Marine Molasse.





For both the MP.M. and Recking approaches we calculated the total bedload sediment flux ($Q_s$) by computing the dimensionless sediment bedload flux. Here, this value is defined by the Einstein parameter ($\Phi$)
(Einstein, 1950):

$$\phi = \frac{Q_s}{W \cdot \sqrt{g \cdot (\rho_s / \rho_w - 1) \cdot D_{50}^3}} \quad (1)$$

Where $Q_s$ is the total bedload sediment flux (m³/s), $W$ is the active river width (m), $g$ is the gravity acceleration (m²/s), $\rho_s$ and $\rho_w$ (kg/m³) are the densities of sediment and water, respectively, and $D_{50}$ (m) is the
50th percentile of the riverbed surface grain sizes (b-axis) and here represents the characteristic diameter of the transported material. In the following, we present the formulation to calculate $\Phi$ using the MP.M. and Recking
approaches.

### 3.2.1 Dimensionless sediment bedload flux ($\Phi$) based on MP.M. approach

For an alluvial stream where water flow is considered uniform, the Einstein parameter ($\Phi$) can be calculated by the formulation proposed by Wong and Parker, 2006:
$$\phi = A \cdot (\tau^* - \tau_c^*)^{1.5} \quad (2)$$

Where $A$ is a non-dimensional constant, which was set to 3.97 by Wong and Parker (2006) based on a

reanalysis of the dataset obtained by Meyer-Peter and Müller (1948) through flume experiments. In this equation, the difference between the dimensionless shear stress ($\tau^*$) and the Shields (1936) parameter ($\tau_c^*$) is
key for estimating how much sediment a stream can entrain from the riverbed if $\tau^* \geq \tau_c^*$. For a non-uniform mixture of grains on a riverbed, the choice of the $D_{50}$ in Eq. 1 is justified for near-equal mobility conditions
when grains larger and smaller than the $D_{50}$ are mobilised at nearly the same rate and for the same shear stress (Julien, 2010). While Wong and Parker (2006) considered a constant value $\tau_c^* = 0.0495$ for the Shields number,
Lamb et al. (2008) proposed to employ a slope-dependent correction, mainly because the consideration of a constant Shields number will overpredict the bedload discharge in Eq. 1 for steep gradients (> 0.02), which is
the case for the Guerbe River. Therefore, we considered the Shields parameter ($\tau_c^*$) to be dependent on the channel bed gradient $S$ (in meter per meter) following the results of field and laboratory experiments (Lamb et
al., 2008):

$$\tau_c^* = 0.15 \cdot S^{0.25} \quad (3)$$

The dimensionless shear stress ($\tau^*$) is defined following Shields (1936):

$$\tau^* = \frac{\tau}{g \cdot (\rho_s - \rho_w) \cdot D_{50}} \quad (4)$$

The shear stress ($\tau$) in a river bed is controlled by the channel depth $d$ (in meters) for streams where the river width is $W > 20\,d$:



$$\tau = \rho_w \cdot g \cdot d \cdot S \qquad (5)$$


Here we considered that the channel has a rectangular configuration. In the case of a uniform flow, the

friction slope can be considered as being identical to the alluvial riverbed slope ($S = S_{bed}$). The water depth is

calculated from the relationship between the unit water discharge ($q = Q \cdot W^{-1}$; m$^2$ s$^{-1}$) and the mean water

velocity along the river depth $v$ (expressed in meters per second):

$$d = \frac{q}{v} \qquad (6)$$

Ferguson (2007) proposed that in a stream, the mean water velocity ($v$) of a water column can be

calculated separately for shallow- and deep-water conditions thereby using the Manning Strickler friction law

and a roughness layer (MS/RL) term:

$$v_d = \frac{a_1^{0.6} \cdot g^{0.3} \cdot S^{0.3} \cdot q^{0.4}}{D_{84}^{0.1}} \qquad \text{(deep flows)} \qquad (7.1),$$

and

$$v_s = \frac{a_2^{04} \cdot g^{0.2} \cdot S^{0.2} \cdot q^{0.6}}{D_{84}^{0.4}} \qquad \text{(shallow flows)} \qquad (7.2)$$

Here $a_1$ and $a_2$ are empirically obtained values and set to 5.5 and 2.5 (Ferguson, 2007), and the $D_{84}$ is

the 84th percentile of the riverbed grain sizes (b-axis). The water column is considered as "shallow" if $d / D_{84} <$

$4$. This formula has the advantage that it can be applied to rivers with a large range of slopes, including those

encountered in mountainous streams where the slopes are steep (Zimmermann, 2010).

**3.2.2 Dimensionless sediment bedload flux (Φ) based on the empirically calibrated Recking approach**


A further method for calculating the reach-average bedload flux for gravelly rivers was proposed by

Recking (2013) under the condition that no morphological changes occur on the channel bed during the

transport of sediment. This means the method can be applied under the conditions that neither the slope nor the

width of the channel bed changes. The related equations were empirically adjusted using a large dataset

collected in the field, and they were validated by blind tests, which were conducted in 15 river reaches.

According to this author, the Einstein parameter (Φ) can be calculated through:

$$\phi = \frac{14 \, \tau^{*2.5}}{1 + \left(\frac{\tau_m^*}{\tau^*}\right)^4} \qquad (8)$$

Where $\tau^*$ is the dimensionless shear stress defined in Eq. 4. The parameter $\tau_m^*$ accounts for the

transition from the situation where only a fraction of the channel bed material is transported (partial transport) to



the condition where all sedimentary material is in transport (full mobility). The original formula presented in 2013 was subsequently updated by Recking et al. (2016) to account for streams with flatbeds and steep-pool
patterns:

$$\tau_m{*} \ = \ 1.5 \cdot S^{0.75} \tag{9}$$

In Eq. 8 the dimensionless shear stress $\tau^*$ is defined by Eq. 4, which is dependent on the flow depth ($d$) to estimate the shear stress (Eq. 5). In the following, we calculated the flow depth using the equation derived by
Recking et al. (2016), which itself bases on the flow resistance formula proposed by Rickenmann and Recking (2011):
$$d \ = \ 0.015 \cdot D_{84} \frac{q^{*2p}}{p^{2.5}} \tag{10}$$

where $q^* \ = \ q/\sqrt{g \cdot S \cdot D_{84}^3}$ and $p \ = \ 0.24$ if $q^* \ < \ 100$, else $p \ = \ 0.31$. Therefore, we re-calculated the

dimensionless shear stress in the following way:

$$\tau^* \ = \ \frac{0.015 \cdot q^{2p} \cdot D_{84}^{1-3p} \cdot S^{1-p}}{p^{2.5} \cdot g^p \cdot (\rho_s/\rho_w - 1) \cdot D_{50}} \tag{11}$$

Piton and Recking (2017) used the Recking et al. (2016) formula to calculate the bedload flux considering different states of armouring on the channel bed and various sources of sediment. They compared
the suitability of the equation to predict the bedload flux by using two different values as the characteristic diameter of the transported material: the 84th grain size percentile of the bedload material in transport labelled
as $D_{84,TraBL}$ and the 84th percentile of the riverbed surface ($D_{84}$ as in Recking et al., 2016), instead of the 50th percentile of the sediments on the riverbed surface ($D_{50}$ in Eq. 11). They concluded that the choice of the
characteristic diameter depends on the geomorphological context of the stream. In particular, for a "colluvial" stream pattern, as is the case for the Guerbe River, the use of the $D_{84,TraBL}$ yielded better model predictions than
the $D_{84}$. Since in our work, we can only measure the grain size distribution representing the riverbed surface, we considered the $D_{50}$ as representing the $D_{84,TraBL}$. We propose that this assumption is acceptable for the Guerbe
River since streams with a "colluvial" pattern are characterized by similar $D_{50}$ and $D_{84,TraBL}$ values (see Fig. 4 in Rickenmann and Fritschi, 2010 for the Erlenbach stream in the Swiss Alps and Fig. 7 in Piton and Recking,
2017 for the Upper Roize stream).

    In summary, both the Meyer-Peter and Müller (1948), here referred to as MP.M, and Recking et al.

(2016) formulations require the same key parameters to calculate the transported bedload, which are: the alluvial slope, the $D_{50}$, and $D_{84}$ grain size percentiles, the channel width, and water discharge.



### 3.3. Data acquisition

**3.3.1. Uncrewed aerial vehicle (UAV) surveys and photogrammetry processing**

We applied a UAV close-range setup in August-September 2021 to measure grain sizes on emerged

gravel bars along the Guerbe River (Figs. 1b and 1c). We designed our surveys (13) and photogrammetric
processing based on the workflow of Mair et al. (2022) with the aim of reducing the uncertainties related to the

survey in the field and the processing of the data on the resulting grain sizes. To ensure a sufficient ground
sampling distance of < 2 mm/pix in all pictures, we conducted close-range surveys with a nominal flight altitude

between 5 and 9 m above ground. For image acquisition, we used a one-level grid of nadir camera positions as
backbone geometry, for which we targeted a lateral and frontal overlap between individual images of 80%. We

complemented this grid with images (5 to 20 per site) taken with oblique angles with a pitch of >20˚. The
images were taken at the same survey altitude in an effort to minimize systematic errors during the

photogrammetric processing (James et al. 2020; Carbonneaun and Dietrich, 2017; James and Robinson, 2014).
All images were taken in the JPEG format with a DJI Mavic 2 Pro on-board camera (Hasselblad L1D-20c),

which utilizes a global shutter. For referencing, we distributed 5 to 10 ground control points (GCPs) over each
target gravel bars and measured them with a Leica Zeno GG04 Plus GNSS antenna with the real-time online

Swipos-GIS/GEO RTK correction. This setup yields a horizontal precision of 2 cm and a vertical precision of 4
cm ($2\sigma$) under ideal conditions (Swisstopo, 2022). The subsequent photogrammetric processing followed

standard structure from motion (SfM) workflows (e.g., James and Robson, 2012; Fonstad et al., 2013, Eltner et
al., 2016) including recent updates (e.g., James et al, 2017a, b; 2020) to produce high quality orthomosaic and

digital surface models (DSMs) for each gravel bar (e.g. Fig. S1). To do so, we used the Agisoft Metashape (v1.6
Pro) software, licensed to the Institute of Geological Sciences, University of Bern. In total, we processed 13

SfM models, with average checkpoint/GCP precision of 26.69 ± 17.72 mm and systematic errors < 10 cm
(Table S1).

**3.3.2. Grain size measurements**

We manually measured the size of grains on the orthomosaics that resulted from the field surveys (see

section above) by applying the approach of Woman (1954). Here we used the QGIS 3.22 open-source software
to create a grid with a 0.5 m-wide spacing and to measure the sizes of grains. For each grain underneath a grid

intersection, we measured the lengths of the a- and b-axes by fixing four dots at the grains' edges, thereby using
these to define the two perpendicular axes (e.g. Fig. S1 and S2 in appendix). Because of the limited resolution of

the images (Table S1 for image resolution), we defined a grain size measurement threshold of 2 cm.
Accordingly, all grains smaller than this threshold were considered as equal to 2 cm. This consideration had no

effect on our values of 50th or 84th grain size percentiles since the proportion of grains smaller than 2 cm was
never larger than 25%. We then calculated the 50th and 84th percentile values from the grain size dataset to

characterize each gravel bar. Following Mair et al. (2022), we estimated the related 95% confidence intervals
using a combined bootstrapping and Monte Carlo modelling approach for which we used the survey-specific

SfM uncertainties (Table S1). Here, we assumed that the grains on the gravel bars are characteristic of the





material that was transported during equal mobility conditions since during these events the surveyed bars were
immersed.

### 3.3.3. Topographic gradients and river widths

298        In the Guerbe River, the transport of the bedload is currently conditioned by the values of the
engineered slopes ($S_c$ in Fig. 3), which we measured from the DSMs obtained from the UAV images (Section
3.3.1). For non-engineered conditions, we inferred that the corresponding slopes ($S_n$ in Fig. 3) would have been
similar to the gradient of a long reach around the site of interest where grain size data was collected (150 m
upstream and 150m downstream), considering an elevation difference between at least 6 check dams. Here we
used the LIDAR DEM swissALTI3D (swisstopo, 2019) with a spatial resolution of 0.5 m$^2$ as a basis. The slope
values were then calculated by taking the difference in the topography of two points in the water flow direction
and dividing this value by the distance between them. For each survey site, we repeated such measurements at
least 30 times to calculate the 95% confidence interval of the slope. Also at these sites, we measured the active
river's width on orthoimages (SWISSIMAGE, spatial resolution of 10 cm; swisstopo) . We determined the
cross-sectional stream widths by measuring the width of the check dams' spillways downstream of our survey
reach, which is considered to represent the engineered river width during flood stages.

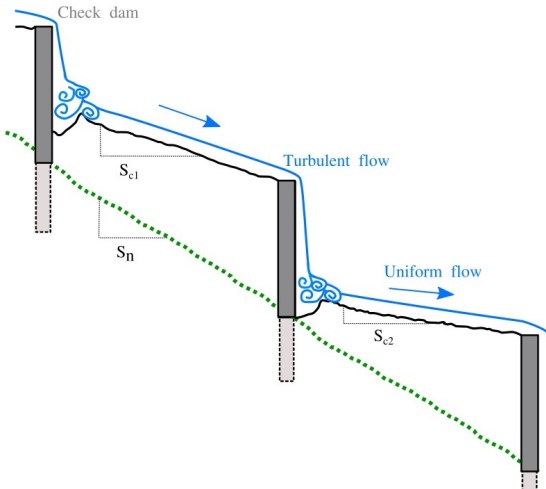

**Figure 3.** Topographic gradient in a reach between check dams: From the engineered riverbed (black line) we
calculated the engineered slope of the river ($S_c$). Likewise, we calculated the non-engineered slope ($S_n$, green
dashed line) using a 300 m-long reach around the site of interest.





### 3.3.4. Surface runoff


The water discharge is a further key parameter for calculating the sediment bedload flux (Eq. 1 to 11).
The runoff can greatly vary over a short time interval, and such variations are even stronger during the
entrainment of sediment particles in mountain streams (Tuset et al., 2016). This implies that information on the
local runoff is necessary to properly calculate the rates of bedload transport. Here, we used the gauging records
at Burgistein (Fig. 1a) as a reference, where sensors have measured the water levels every minute since 2009.
These values have then been converted to water discharge based on an empirical relationship in which the
related parameters were acquired at Burgistein (Spreafico and Weingartner, 2005). This station has been
operated by the Bau- und Verkehrsdirektion des Kantons Bern (https://www.bvd.be.ch/), which offered us the
water discharge data acquired between 2009 to 2021.


Since our area of interest is situated upstream of the Burgstein station (Fig. 1a), we downscaled the
runoff values measured at Burgistein ($Q_b$) for our sites of interest ($Q_l$) by a factor that depends on the ratio
between the size of the upstream catchment of the selected site ($A_l$) and that of the Burgistein station ($A_b$). This
value was then multiplied by the ratio between the mean annual precipitation rate for the corresponding
catchment contributing to water runoff at the selected site ($P_l$) and the Burgistein station ($P_b$):

$$Q_l = \frac{A_l}{A_b} \cdot \frac{P_l}{P_b} \cdot Q_b \qquad (12)$$


Here, we employed an annual precipitation rate value of $P_l$ = 1734 mm for our study reach and $P_b$ =
1492 mm for the basin (Ramirez et al., 2022), which contributes to the runoff at Burgistein. We then used the
gauging data collected over the past 12 years, based on which we estimated the range of bedload flux and also
the total volume of sediment transported during this time, and we did so for engineered and non-engineered
conditions in the Guerbe River. We acknowledge that the estimation of runoff upstream of a gauging station
depends on multiple factors such as the groundwater level, the type of vegetation, and the thickness of the soil
(Sriwongsitanon and Taesombat, 2011). However, since our gauging station is only c. 4 km downstream of our
area of interest, we inferred that neglecting these factors will not significantly bias our estimations of the local
runoff values.

### 3.3.5. Propagation of uncertainties in estimating the bedload flux


We applied a Monte Carlo simulation to estimate the uncertainties of the bedload predictions. We
proceed through using the uncertainties that occur upon measuring the values of the key variables as input
parameters, and through fitting the gamma distributions for the range of uncertainties that are associated with
the percentiles of the grain size datasets (i.e., the 95% CI on the $D_{50}$ and $D_{84}$). These were obtained with the
method proposed by Mair et al. (2022) to simulate the related uncertainties. The scale and shape parameters of
the gamma distributions that we employed for the Monte Carlo simulation are presented in Table S3. We used
normal distributions for all engineered and non-engineered slopes, with the standard deviation calculated from
the 95% confidence interval divided by 4 . For estimating the uncertainties on the width values we applied a





uniform distribution where the length of this distribution was defined using the measured width including a ±

10% uncertainty at each site.

**3.4. Considerations of non-engineered scenarios**

For the non-engineered scenarios, we considered changes not only in the slope but also in the river width and grain sizes. In particular, in a natural state, the channel widths are expected to be smaller than the
widths of check dams' spillways as is currently the case. This has been shown in various engineered mountainous streams (Piton et al., 2017; Lucas-Borja et al., 2021) and is likely also valid for the Guerbe torrent.
However, predictions of natural channel widths can be challenging because the hillslope instabilities around the channel can strongly affect this parameter, and information on widths was not available for the time before the
check dams in the Guerbe River were constructed. Therefore, we had to make assumptions and considered three scenarios in which the current widths were shortened by 75%, 50% and 25%. Although we lack constraints to
sustain these inferences, we justify the selection of these values because upstream of site 1 where the Guerbe River is poorly engineered, the channel widths are generally narrower than the width values we get when
applying a 50% shortening. In the same sense, a prediction of grain size patterns for non-engineered conditions is speculative because of a lack of observations. Here, we used the grain size values from the bulk material
upstream of site 1, which we considered as characterizing the source signal. Indeed, mapping shows that the highly active hillslopes just upstream of site 1 have most likely been the primary material source (Figs. 1a and
2). Furthermore, because riverbed grain sizes can also be affected by abrasion during transport in mountainous torrents (Miller et al., 2014), predictions about how the calibre of the bedload material changes downstream are
almost impossible to make particularly for non-engineered states in the past. Therefore, we considered the grain sizes of the inferred supply signal as maximum values, which we kept as a constant parameter along the
surveyed sites for some scenarios. Consequently, the non-engineered scenarios presented in this work will base on conservative assignments of values to the parameters, which control the transport of bedload material.
**4. Results**

**4.1 Grain size, channel slope and width, and water discharge**

We obtained data on grain sizes of sedimentary particles on the riverbed surface and channel slopes for engineered and non-engineered conditions for all 13 surveyed sites (Fig. 4). The $D_{50}$ values resulting from the
measurements show a decreasing trend from c. 8.3 cm to 2.4 cm in the downstream direction (Fig. 4a). In contrast, the sizes of the $D_{84}$ rapidly decay between sites 1 and 2 from > 25 cm to < 20 cm, after which the
values fluctuate between c. 20 and 10 cm (Fig. 4b). The measured slopes for engineered conditions display a similar pattern as the $D_{84}$ in the sense that the energy gradient rapidly decreases from c. 10 to 5 cm m$^{-1}$ between
sites 1 and 2. The gradients then oscillate around a value of c. 3 cm m$^{-1}$ farther downstream (Fig. 4c). This pattern of alternating slope values is clearly visible for the reaches between all check dams in the dataset
obtained from the 0.5 m SwissAlti3D DEM where the data collection was achieved in 2019 (Fig. S3). The non-engineered slopes are substantially different. They are flattest at site 1 and along the downstream portion of the
fan (from site 7 onwards) where the values are c. 10 cm/m and less (Fig. 4d). In-between, the energy gradients

continuously decrease in the downstream direction, starting with c. 20 cm m$^{-1}$ at site 2 and ending with a value of 10 cm m$^{-1}$ on the fan itself (Fig. 4d). This rapid increase in energy gradient between sites 1 and 2 points to the occurrence of a knickpoint in the longitudinal stream profile (see section 5.3 for more details), which is also corroborated by the geomorphological map where several break-in-slopes are visible on the hillslopes bordering the channel system in this area (Fig. 2). The current channel widths (thus during engineered conditions) fluctuate around a value of 15 m without displaying a clear trend in the downstream direction (Table S2).

The pattern of water discharge along the surveyed reach was calculated using Eq. 12 and the records at the Burgistein gauging station as a basis (Figs. S4). Accordingly, at the fan apex, the peak annual discharge values vary between 5 and 18 m$^3$ s$^{-1}$ (Fig. 5) in which the highest discharge event during the surveyed period occurred in 2021.

**4.2 Bedload flux for engineered and non-engineered scenarios**

We calculated the volumes of the instantaneous and mean annual bedload that can be transported along the surveyed sites by applying the MP.M. and Recking formula. Considering the constraints as elaborated in sections 3.4 and 4.1, the results show that for the engineered conditions, the mean annual bedload transport rate at the fan apex ranges from c. 1'000 to 6'000 m$^3$ yr$^{-1}$ if the MP.M. equation is used, or from 900 to 2'500m$^3$ yr$^{-1}$ if the calculations are done with the Recking approach (Fig. 6). For the non-engineered state, we calculated mean annual transport rates that are between c. 10 (Recking formula) and 100 times higher (MP.M. formula). More specifically, the values for bedload transport at the apex vary from 30'000 to 400'000 m$^3$ yr$^{-1}$ using MP.M.'s equation for all scenarios of channel width shortening and grain sizes (Fig. 6a). Alternatively, the values are smaller if estimated with the Recking equation, and they vary between 1'000 to 1'500 m$^3$ yr$^{-1}$ (Fig. 6b). See a detailed discussion on these differences in section 5.1.

Along the segment upstream of the apex, the mean annual bedload fluxes calculated for all surveyed sites revealed specific patterns both for engineered and non-engineered conditions and also for the MP.M. and Recking approaches (Fig. 7). For the engineered conditions the use of the MP.M. equation predicts the highest bedload flux of c. 10'000 m$^3$ yr$^{-1}$ for site 1, whereas the fluxes are c. 50% lower at the other sites and are approximately 5'000 m$^3$ yr$^{-1}$ (Fig. 7a). In contrast, the application of the Recking equation returns values of mean annual bedload flux that are less than 1'000 m$^3$ yr$^{-1}$ for all sites upstream to the fan apex (Fig. 7c). For the non-engineered conditions, the application of the MP.M. equation shows a rapid increase in the bedload capacity between sites 1 and 2, after which the values fluctuate around c. 400'000 m$^3$ yr$^{-1}$ in the downstream direction until the fan apex (Fig. 7b). In contrast, the application of the Recking approach predicts that sediment flux continuously increases from <1'000 m$^3$ yr$^{-1}$ in the headwaters to >60'000 m$^3$ yr$^{-1}$ near the fan apex (Fig. 7d). If the stream's response to peak discharge conditions is considered, then for engineered conditions the MP.M. equation returns a peak sediment flux at site 1 of 0.3 m$^s$s$^{-1}$, after which the bedload flux fluctuates around a constant value that is c. 3 times lower than at site 1 (Fig. 8a). The pattern is similar if the Recking equation is used, but the values are generally 50% lower (Fig. 8c). In addition, also using the Recking equation, site 1 has a predicted sediment flux that is the same as farther downstream. If the non-engineered states are considered, then



the application of the MP.M and Recking equations show both the same pattern for the peak discharge

scenarios, where the bedload fluxes during peak discharge are between 8 (MP.M equation) and 20 times higher

(Recking equation) than for engineered conditions (Figs. 8b and 8d).

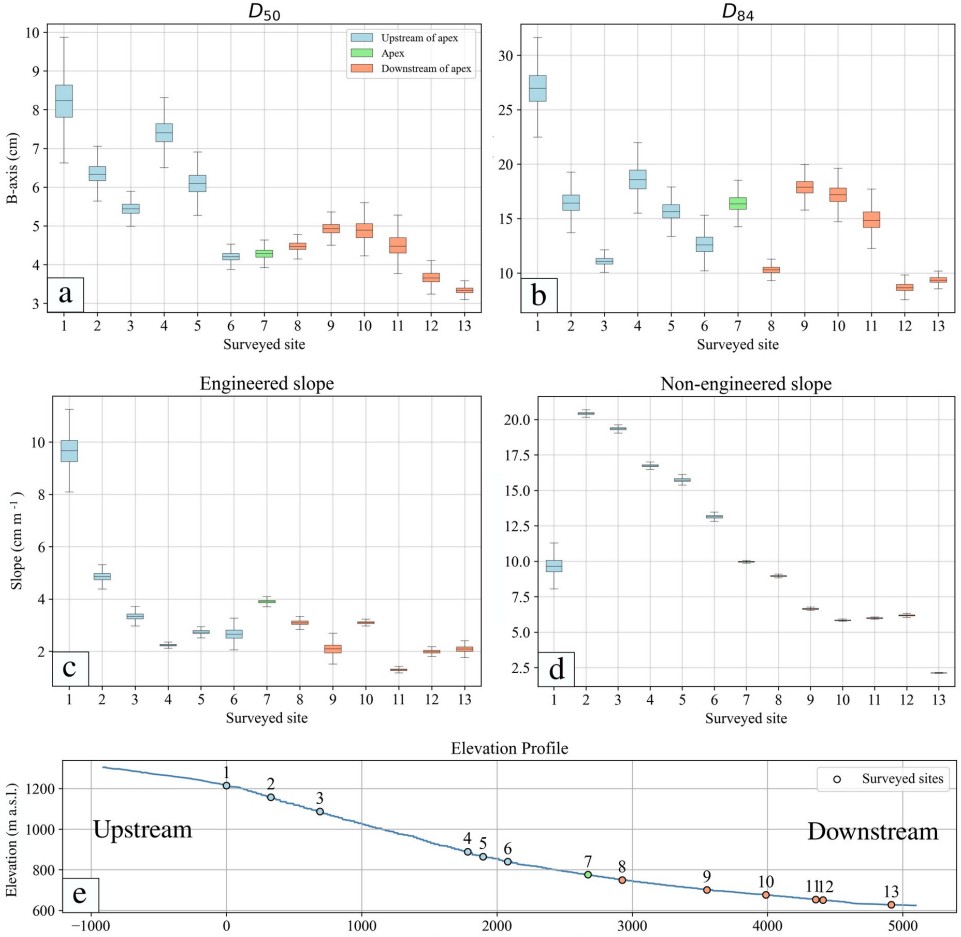

**Figure 4.** Boxplots representing the measured parameters at the surveyed sites with propagated uncertainties: (a)

Bed surface grain size $D_{50}$, (b) size of the $D_{84}$ of the sediments on the bed surface, (c) alluvial slope for the

engineered conditions, (d) alluvial slope obtained from the DEM for non-engineered conditions. (e) Elevation

profile of the Guerbe River. The sites upstream and downstream of the alluvial fan's apex are indicated by the

blue and red colours, respectively, and the site on the apex is indicated by the green colour.



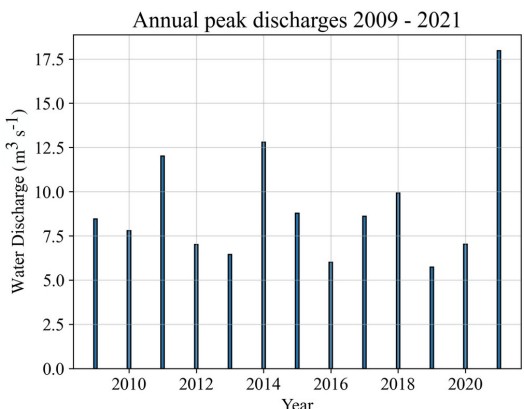

**Figure 5.** Calculated values of annual peak discharge for the fan apex of the alluvial fan in the Guerbe River
during the period between 2009 and 2021.

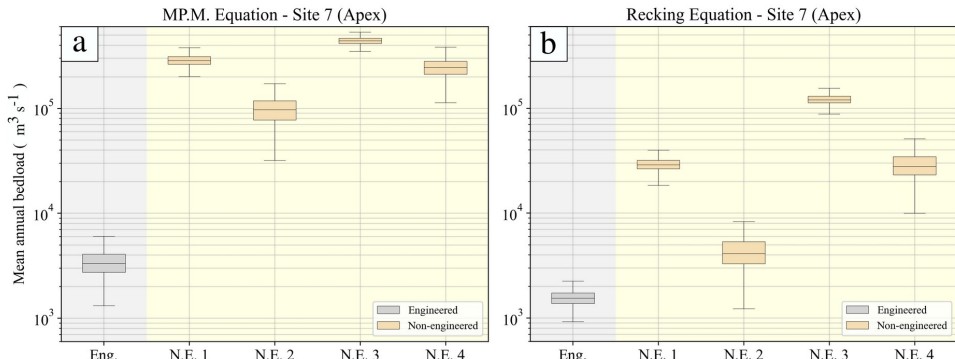

**Figure 6.** Boxplot representation of the mean annual bedload estimates using (a) the MP.M. and (b) the Recking
approaches for the Guerbe River catchment. The engineered (Eng.) and the non-engineered (N.E.) scenarios are
based on using the parameters shown in Fig. 4. Specifically, the engineered scenario is based on the average of
the engineered slopes, whereas the results for the non-engineered scenarios are based on: (N.E.1) a 75%
reduction of the channel width and grain sizes from site 7; (N.E. 2) a 75% reduction of the channel width and
grain sizes from site 1; (N.E. 3) a 25% reduction of the channel width and grain sizes from site 7; and (N.E. 4)
25% reduction of the channel width and grain sizes from site 1.

438          Along the segment upstream of the apex, the mean annual bedload fluxes calculated for all surveyed
sites revealed specific patterns both for engineered and non-engineered conditions and also for the MP.M. and
Recking approaches (Fig. 7). For the engineered conditions the use of the MP.M. equation predicts the highest
bedload flux of c. 10'000 m$^3$ yr$^{-1}$ for site 1, whereas the fluxes are c. 50% lower at the other sites and are
approximately 5'000 m$^3$ yr$^{-1}$ (Fig. 7a). In contrast, the application of the Recking equation returns values of
mean annual bedload flux that are less than 1'000 m$^3$ yr$^{-1}$ for all sites upstream to the fan apex (Fig. 7c). For the





non-engineered conditions, the application of the MP.M. equation shows a rapid increase in the bedload capacity between sites 1 and 2, after which the values fluctuate around c. 400'000 m$^3$ yr$^{-1}$ in the downstream
direction until the fan apex (Fig. 7b). In contrast, the application of the Recking approach predicts that sediment flux continuously increases from <1'000 m$^3$ yr$^{-1}$ in the headwaters to >60'000 m$^3$ yr$^{-1}$ near the fan apex (Fig.
7d). If the stream's response to peak discharge conditions is considered, then for engineered conditions the MP.M. equation returns a peak sediment flux at site 1 of 0.3 m$^3$ s$^{-1}$, after which the bedload flux fluctuates
around a constant value that is c. 3 times lower than at site 1 (Fig. 8a). The pattern is similar if the Recking equation is used, but the values are generally 50% lower (Fig. 8c). In addition, also using the Recking equation,
site 1 has a predicted sediment flux that is the same as farther downstream. If the non-engineered states are considered, then the application of the MP.M and Recking equations show both the same pattern for the peak
discharge scenarios, where the bedload fluxes during peak discharge are between 8 (MP.M equation) and 20 times higher (Recking equation) than for engineered conditions (Figs. 8b and 8d).
Downstream of the apex, both equations yield the same pattern where both the peak and mean annual bedload fluxes have lower values than at the apex (Figs. 7 and 8). Yet, for the engineered conditions, we
observed that the flux pattern locally reached high values particularly if the Recking equation is applied. Finally, we also estimated the locations where a riverbed armour breaking might have occurred during the 2021 peak
discharge using the D$_{84}$ grain size as a threshold in the Shields equation (Eq. 4; see Table S4 and also Schlunegger et al., 2020). When armour breaking occurs, we expect a large amount of material to be transported
and also a change in the morphology of the channel (e.g. slope variations) during and after the event. The results show that such a reorganization of the channel bedform could potentially occur at a few sites only irrespective
of the selection of a particular equation. For a non-engineered situation and using the Recking approach, however, all sites are predicted to experience armour-breaking conditions during a flood with a magnitude that
would correspond to the one during the 2021 peak discharge (Table S4).

**5. Discussion**

The application of two different approaches to calculate the bedload transport capacity revealed specific differences, which become more important when considering the non-engineered status. In contrast,
where bedload transport rates are calculated for engineered conditions, the differences resulting from the two formulations are less and within uncertainties. This will further be discussed in section 5.1. Thereafter, we
discuss how the check dams potentially contribute to the regulation of sediment transport (section 5.2) and how the stabilization of the channel bed affects the consolidation of the hillslopes (section 5.3).

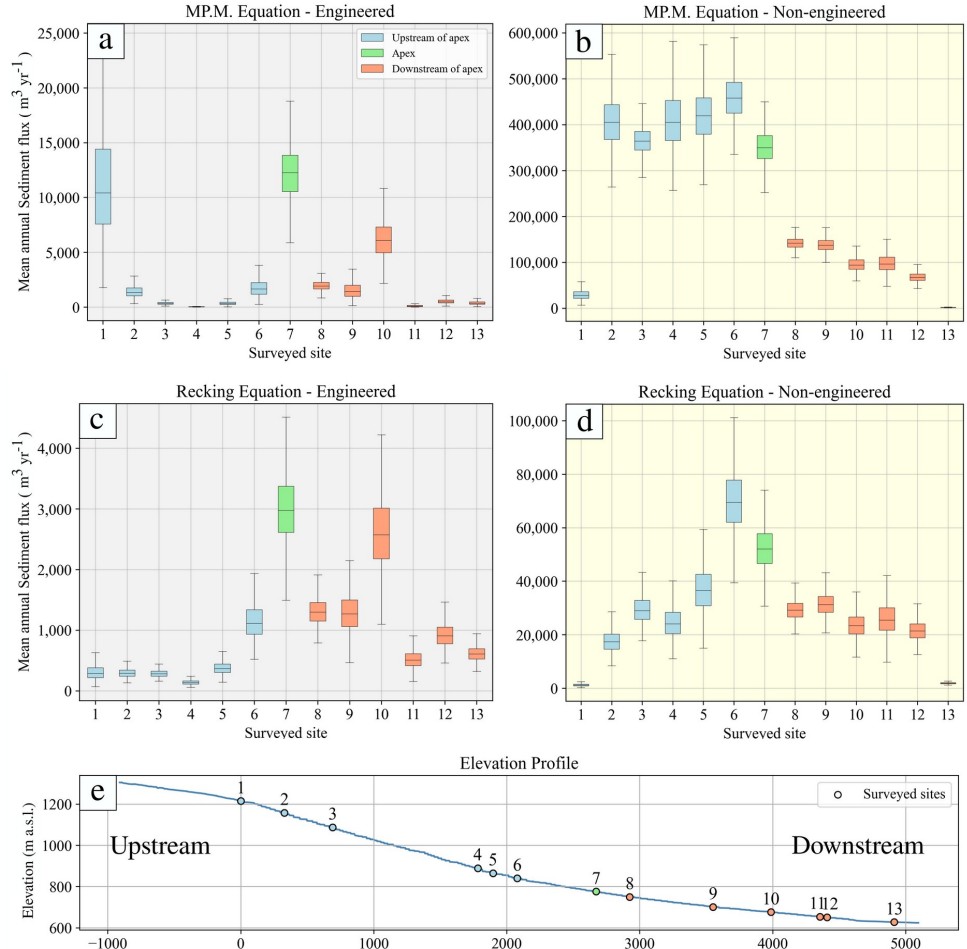

**Figure 7.** Boxplot representation of the annual mean bedload estimates using the MP.M. and the Recking along all the surveyed sites. For the values of the parameters to compute the sediment flux for engineered (a and c) and non-engineered (b and d) scenarios, please refer to Fig. 4. Specifically, the non-engineered scenario is based on the assumption that the width of the channel is reduced by 50%. The sites upstream and downstream of the alluvial fan apex are indicated by the blue and red colours, respectively, and the site on the apex is indicated by the green colour.

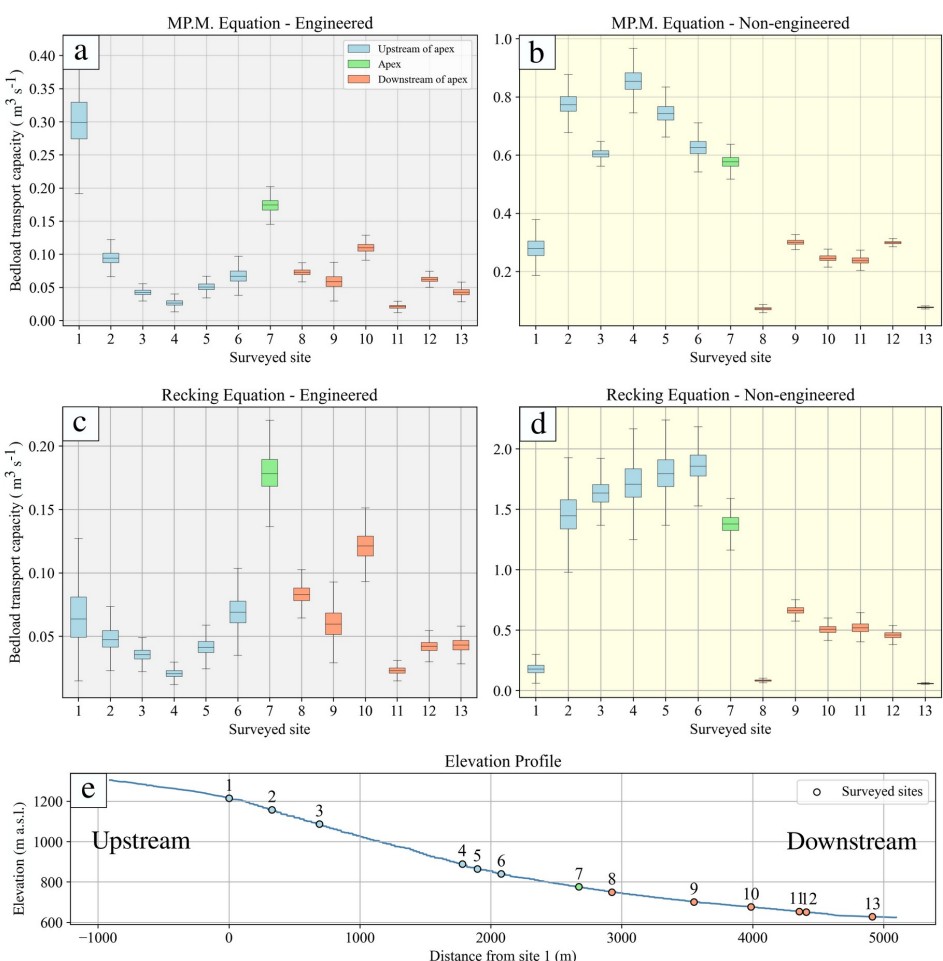

**Figure 8.** Boxplot representation of the bedload predictions using MP.M. and Recking during the 2021 peak runoff along all the surveyed sites. The engineered (a and c) and the non-engineered (b and d) scenarios are based on the parameters shown in Fig. 4. Specifically, in the non-engineered scenario we show the results where 50% of the channel width is employed. The sites upstream and downstream of the alluvial fan apex are indicated by the blue and red colours, respectively, and the site on the apex is indicated by the green colour.

**5.1. Analysis of the equations' results**

Although the MP.M. (Meyer-Peter and Müller, 1948) and Recking (Recking et al., 2016) models have been validated using the results of flume experiments and field surveys (Recking, 2012; Recking et al., 2016), they present strong limitations if they are applied to natural cases where the conditions (e.g., channel slope, discharge values and grain sizes) differ from those with which they were calibrated. In our case, the Guerbe





River has an energy gradient that is up to 10 times steeper for non-engineered conditions than was originally

considered for calibrating the MP.M. equation (Wong and Parker, 2006). The correction by Lamb et al. (2008)

where the Shields variable is modified by a slope-dependent coefficient could account for this problem, but the

results have neither been validated in the field nor through flume experiments. In contrast, the hydro-

geomorphic conditions in the Guerbe River are within the constraints that were used to validate the Recking

equation for both the engineered and non-engineered states (Piton and Recking, 2017). Therefore, the use of the

Recking approach may be more suitable to predict the sediment flux for the situation where the Guerbe River

has no check dams.

For non-engineered conditions, we consider the MP.M. approach to yield a strong overestimation of the

mean annual sediment bedload flux if the results of the Recking equation are taken as a reference. We justify the

selection of this benchmark because the Recking formula was explicitly validated with data from steep

mountainous catchments such as the Guerbe River (see above). This overestimation of the bedload transport

rates mainly concerns the cases of low water discharge (Fig. 9b). Because low water fluxes occur more

frequently during one year than peak discharges, the mean annual bedload transport rates will be higher. For

peak discharges, however, the Recking equation predicts much higher sediment fluxes than the MP.M. equation

(Fig. 9b). Since the Recking approach was also validated for peak water flux (see above), we consider the

resulting values for the Guerbe River as realistic. For the engineered conditions, however, both equations predict

similar sediment fluxes during low and high runoff (Fig. 9a), thereby explaining why predictions of mean

annual sediment fluxes are nearly the same for both equations.

     We also compare our outcomes with two available studies in the Guerbe catchment. The first one

estimated the sediment budget from [10]Be concentrations in the catchment (Delunel et al., 2020), where a

denudation rate of approximately 260 mm kyr$^{-1}$ on our surveyed catchment area gives a mean annual sediment

yield of c. 3'000 m$^3$ yr$^{-1}$. Conventionally, cosmogenic data integrate denudation of times scales of several

thousands of years (von Blanckenburg, 2005) and as such this value would correspond to the total sediment flux

prior to the construction of the check dams. However, as will be argued below, the construction of these steps

resulted in a partial disconnection between the shallow-seated landslides and the Guerbe River particularly

along the margin of the trunk channel (e.g. the Riselbruch landslide which became stabilized after the check

dams were built, see section 5.3). Because the foot of a landslide has been documented to release material with

low [10]Be concentrations (Cruz Nuñes et al., 2015), we anticipate that during pre-engineered conditions the

concentrations of cosmogenic [10]Be in riverine quartz would have been lower. Therefore, we consider the

sediment flux of the c. 3'000 m$^3$ yr$^{-1}$ as representative of the current state. The second study used the CEASAR-

Lisflood evolution model to estimate the total sediment load (suspended and bedload) for engineered conditions,

where a mean annual sediment load of 1'222 m$^3$ yr$^{-1}$ was predicted (Ramirez et al., 2022). Both results can be

converted to mean annual bedload fluxes by applying a 60% factor, based on the results of sediment budgets

carried out on mountain streams in the Alps for basins that are c. 10 km$^2$ large (Schlunegger and Hinderer,

2003). Therefore, applying these corrections for the current engineered state, the [10]Be-based bedload flux is c.

1'800 m$^3$ yr$^{-1}$, whereas the related value derived with the CEASAR-Lisflood evolution model would be in the

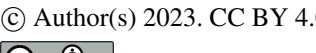



range of c. 700 m$^3$ yr$^{-1}$. Considering the uncertainties that are associated with estimating bedload transport, the

cosmo-based sediment flux and the estimates by Ramirez et al. (2022) are in agreement with the outcome of our

calculations based on the Recking formula.

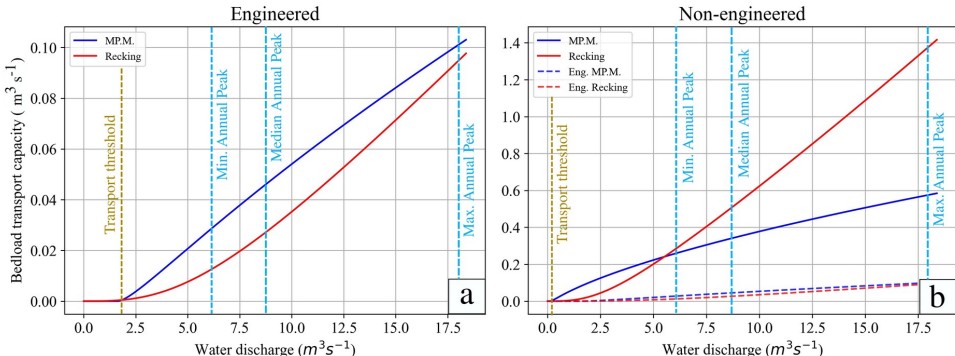


**Figure 9.** Predicted bedload versus water discharge  patterns using the MP.M and Recking approaches for (a)

engineered and (b) non-engineered conditions. These patterns were determined using data collected at site 7

(Fig. 4). Specifically, shown in this figure, the results for the engineered scenario are based on the average of the

engineered slopes, and those of the non-engineered scenario considered a 50% reduction of the channel width at

site 7, and the grain size data that was also collected at that site.

**5.2. Regulation of sediment transport**

For engineered conditions and considering the last peak water discharge event in 2021, the predictions

using the MP.M. and Recking approaches reveal site-specific fluctuations in both the transport capacity and the

armour-breaking probability (Fig. 8 and Table S4). This pattern suggests that sediment transport is regulated

through buffering effects where during a peak discharge event some sites will store a fraction of the supplied

sediment while others will release a large portion of the previously stored material. Such regulation has already

been described for filled check dams where the concrete structures (such as check dams) create fixed points

along a longitudinal profile of a river, which disconnects the reaches between the dams (Piton et al., 2017). In

addition, check dams reduce the length of the reach where spontaneous erosion could occur, thereby reducing

the risk where large volumes of sediment are released and transported downstream in a short time (Piton and

Recking, 2016). We consider that the occurrence of such a regulation is recorded by the downstream

fluctuations of the alluvial slopes (Figure 4 and S3) where segments with flat slopes have the potential to store

further material, whereas reaches with steep slopes will likely represent a sediment source during a next event

when large water fluxes occur. As an additional consequence of such a regulation, the grain size will rapidly

fine downstream through selective transport, particularly along the depositional sites. Such a mechanism was

predicted by theory (Paola et al., 1992) and is documented by our data (Fig. 4). Note, however, that besides

selective transport, the breaking of grains as they fall from the dams into the pool likely also contributes to the

fining of the material (Miller et al., 2014).



### 5.3. Bed stabilization and hillslope consolidation

We interpret that the check dams contribute effectively to the bed stabilization of the Guerbe River (Piton et al., 2017; Lucas-Borja et al., 2021). We infer the occurrence of such a mechanism at work using the results of the MP.M. and Recking equations, both of which predict that in the absence of check dams, the mean annual transport capacity would be substantially higher. This is particularly true along the segment between sites 1 and 2 when the predictions of the sediment flux for the non-engineered state is compared to the flux values characterizing the engineered conditions (Fig. 7b and 7d). This is also the region where we mapped a major knickzone on the hillslopes that border the channel network (Fig. 2). Such features are usually considered as evidence for the occurrence of high surface erosion and sediment production rates (Whittaker and Boulton, 2012; Van den Berg and Schlunegger, 2012; Battista et al., 2020), and they would most likely represent the sites of major sediment production in the case that no check dams had been built. It appears that the check dams are stabilizing the bed, thereby reducing the erosional potential along the reach, which otherwise would be an important sediment factory.

Were it the case that the Guerbe riverbed was not stabilized, then fluvial erosion could lead to an increase in sediment supply through activating shallow-seated landslides (Piton et al., 2017; Lucas-Borja et al., 2021). Such a mechanism at work has been documented for the Erlenbach River, which is an Alpine torrent in Central Switzerland (Rickenmann and Fritschi, 2010). For this basin, Molnar et al. (2010) documented an increase in the slip rates of landslides following a period of rapid fluvial dissection. For the case of the Guerbe basin, an inspection of satellite images taken between 1970 and the present from the Guerbe River discloses that between sites 1 and 2 the landslide activity in the Riselbruch (Knickpoint zone in Fig. 2, S5a and S5b) decreased after the construction of the check dams along this reach leading to a reforestation of the area (Fig. S5c and S5d). We use this example to argue that the check dams in the Guerbe River contribute to the consolidation of the hillslopes (Piton et al., 2017; Lucas-Borja et al., 2021). This mechanism results in a stabilisation of the terrain surrounding the channel, which allows the growth of a stable vegetation as the landsliding activities decrease. Furthermore, the application of the Recking equation predicts that in the absence of check dams, such a hillslope de-consolidation will not only occur in the uppermost area surrounding the knickzone but also along the entire reach upstream of the fan apex (Fig. 7d). We base this inference on the predicted downstream increase in the bedload sediment flux.

### 5.4. Are check dams really effective in reducing hazard impact?

From our results, we conclude that the presence of check dams in the Guerbe River does reduce the bedload flux outcoming from the sediment production area, thus reducing the potential for hazards in the downstream reaches of the stream. However, this conclusion is only valid if we assume that the check dams will not fail over time, which has indeed not been the case with the Guerbe River during the past hundreds of years (Salvisberg, 2017). In fact, Ramirez et al. (2022) showed that a failure of one or multiple check dams releases a large amount of the material that was originally stored behind the concrete structures. These authors also showed that such failure can initiate a cascade where other dams will break in the downstream direction. It is



possible that re-activations of deep-seated landslides can initiate such a failure. Recently, the displacement of the deep-seated Meierisli landslide has damaged >10 of these structures (Andres and Badoux, 2019), with the
consequence that some of them are likely to break and thus to fail in the next years. It was also found by the responsible engineers (G. Hunziker, pers. comm. 2022) that the slip of such landslides has not been influenced
by the presence of check dams during the past decades, with the consequence that they have constantly applied lateral stress on the concrete structure, causing them to eventually break. Consequently, in order to guarantee the
functioning of the check dams as we described above, it is necessary that such infrastructure will be continuously maintained and repaired after some damages, and that the deep landslides will eventually be
surveyed and engineered if possible. From a broader perspective, the results of our study can be extended to other steep mountain streams that have already been managed with such infrastructure. In addition, we propose
that the outcome of our analysis might be used as guidelines for projects that aim at building a staircase system along a steep mountainous stream.
**6. Conclusions**

The analysis presented above shows that the current presence of check dams in a steep alpine stream

(Guerbe River) has a major influence on mitigating the sediment production in the catchment and, consequently, reducing the risks of hazards related to high sediment fluxes. We applied two different approaches to calculate
bedload fluxes, which were based on the Meyer-Peter and Müller (MP.M.) and the Recking equation, and we applied them for engineered and non-engineered conditions. Both equations resulted in similar predictions
regarding mean annual bedload fluxes for the currently engineered state. In contrast, models that are based on the Recking solution predict an increase in bedload flux for non-engineered conditions that is c. 10 times higher
than for the engineered state, whereas the MP.M. equation predicts a bedload flux that is even 100 times larger. Since the Recking approach was calibrated with data from mountain streams with a channel floor morphology
characterized by steps and pools, we consider the resulting predictions for non-engineered scenarios as more reliable than those derived from the MP.M. formula. Importantly, we find that the check dams regulate sediment
transport through buffering pulses of sediment during high discharge conditions. In particular, reaches separated by check dams can either function as a sedimentary sink or as a material source. This is observed by the
downstream variations of local energy gradients where segments with a higher slope could potentially act as a sediment source, whereas reaches with flatter slopes have the potential to store some of the supplied material.
As a second function, we considered that check dams contributed to the stabilization of the channel bed. We infer this by our model results, particularly for the uppermost region where check dams were built. There, for
non-engineered conditions, the models predict a large increase in the bedload transport rate where the slope rapidly increases downstream of a knickpoint, as would be expected for a reach characterized by a knickpoint
retreat. For engineered conditions, however, our models predict that the transport rates of bedload material remain stable despite the occurrence of a knickpoint. As a consequence, the retreat of this particular knickpoint
will not occur as long as the check dams are in operation. Finally, we infer that check dams also contribute to the stabilization of the bordering hillslopes, mainly because they prevent the stream from incising into the



substratum. Therefore, we conclude that our approach is a useful and promising tool to evaluate the first-order efficiency of check dams in reducing bedload sediment flux in steep mountain streams.
**Notation**

The following symbols are used in this paper:

$A$          catchment area (m²);

$D_{50}$      sediment diameter such that 50 % of the bed surface mixture is finer grained (m);

$D_{84}$      sediment diameter such that 84 % of the bed surface mixture is finer grained (m);

$\Phi$          dimensionless Einstein parameter;

$g$          gravity acceleration (m s⁻²)

$Q_s$       bedload (m³ s⁻¹);

$q$          unit water discharge (m² s⁻¹);

$Q$          water discharge (m³ s⁻¹);

$\rho_s$        sediment density (2600 kg m⁻³);

$\rho_w$        water density (1000 kg m⁻³);

$S$          Energy slope (m m⁻¹)

$\tau^*$        dimensionless shear stress;

$\tau_c^*$       Shields number (dimensionless);

$\tau_m^*$      Recking equation parameter (dimensionless);

$\tau$          shear stress (N m⁻²);

$v$          mean water velocity in depth (m s⁻¹)

$W$          channel width (m);

*Acknowledgments.* The authors are grateful for the Bau- und Verkehrsdirektion des Kantons Bern
(https://www.bvd.be.ch/), which kindly offered us the water discharge for the Guerbe River and the SWISSTOPO for the offered DEMs and images.
*Financial support.* This research has been supported by the University of Bern and Innovative Training Network S2S (grant no. 860383).



*Author contributions.* AHdP, DM, PG and FS applied the UAV close-range setup in the Guerbe River. DM designed the UAV close-range setup applied in this work. AHdP measured the grain sizes and performed the analyses described in this paper. AHdP wrote the manuscript with support from DM, PG, CS, ACW, SC and FS.

*Competing interests.* The contact author has declared that neither they nor their co-authors have any competing interests.

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
