# Peer review of "Check dam impact on sediment loads: example of the Guerbe"

_Hydrology and Earth System Sciences, 2023_

## Author Response (AR1)

This article aims to evaluate the impact of the presence of check dams on bedload transport in mountain streams. This question is crucial and recurring from the operator in charge of this structures, because it is very costly. The methodology used in the paper consisted of calculating and comparing the transport dynamics in the Guerbe river, before and after the construction of the check dam, using bedload equations.

I found the article well written, the data used are of good quality and the results interesting. The weakness probably lies in the use of bedload equations which are very uncertain. One way to counter possible criticism would perhaps be to place more emphasis on the fact that the equations are considered here in a relative manner (before and after construction). In the same vein, the choice of the two equations could also be justified by the fact that they represent to end members in the large panel of available equations (flume derived equation with tendency to overestimate versus field derived equation with low estimate).

I suggest major revision because it is necessary to reconsider some paragraphs, but nothing insurmountable.

We appreciate your insightful review and recognition of this study. Your constructive comments will be valuable in enhancing our manuscript during the review process.

We recognize that the equations to estimate the bedload fluxes present substantial uncertainties, which can be of ± 1 order of magnitude when considering the predictions for single events (Recking, 2013, see reference in the main text). Nevertheless, the available equations are calibrated to estimate average bedload fluxes by taking into account multiple events with identical conditions of water discharge, slope, and river width. This makes them a potent tool for a study focusing on relative changes in sediment transport capacity within a specific catchment. In fact, the main idea of this study is to show the relative differences in these estimations for engineered and non-engineered conditions using two end members of the available equations. We will enhance the clarity of this objective in the revised manuscript.

COMMENTS

Line 10 : « human » not necessary ?

We agree it is not necessary. The term "human" has been removed from the text.

Line 17 : I do not agree. You actually apply one method with 2 equations.

We apply one method for data acquisition and 2 different equations to estimate the bedload. We reformulated this statement in the text.

Line 19 : same remark : reformulate

Suggestion accepted.

Line 33 : I agree that the width was probably smaller before the dam construction, but I do not really understand the argument proposed here

Here we argue that a lateral supply of material from landslides could also make the channel even more narrow without the presence of checkdams. This statement is not necessary for the abstract, therefore it has been removed.

Line 51: Piton et al 2017 also discuss a third reason which is a buffering effect (in abstract : "buffering effect, releasing frequently and in small doses what, in their absence, would be transported abruptly en-masse during rare extreme events)

Thank you for this suggestion. We call this effect "Regulation of the sediment transport" in section 5.2. The description of this effect was added to the introduction of the manuscript (Line 55).

Line 73: it is a general comment: maybe not necessary to cite equation names here

The names were removed from this phrase.

Line 129: I think there is a paragraph 3.0 missing where you explain the philosophy of the method in more detail. You just mentioned in the introduction that you will use equations but this is questionable because we all know the uncertainties associated with equations. I think you should start by recognizing this strong limitation, and explaining that you approach it by considering not the equation as a truth but that what interests you is rather their relative results (before and after).

Thank you for this suggestion. We added a new paragraph (Lines 130-149) in the Methods section to emphasize our acknowledgement of the high uncertainties when applying the equations to predict bedload fluxes for single events. However, our focus lies in the average relative variation of the bedload flux, not the absolute count of single events, as this helps evaluate the efficiency of the check dams. We also highlight that both equations provided estimations of the same order of magnitude for the annual mean bedload under engineered conditions, aligning with estimations from other methods in the same catchment (See more in section 5.1). The primary distinction between the equations becomes evident under steeper non-engineered conditions.

Line 148: It must first be recognized that numerous equations have been proposed in the literature, including for steep slopes. The choice is of course difficult but an argument could be to test these two equations chosen at the ends of the wide range of equations available: the flume equation of MPM knows to have a tendency to overestimate and that of Recking derived from the field and generally calculate less transport compared to channel equations.

We rephrased the introduction of this section (Line 172-190). The primary objective is to justify the selection of two equations, representing the wide range of available equations: flume experiments-based equations and field-based equations. Choosing a representative flume-based equation for our case study is challenging because available equations tend to overestimate bedload fluxes for our specific range of grain sizes and slopes. Hence, we opted for the MP.M. equation due to its simplicity and suitability for the gravel grain size domain. Additionally, we incorporate a correction for the critical Shields shear stress along river reaches with slopes steeper than $0.02$ m m$^{-1}$. For the field-based equations, we chose the equation proposed by Recking (2013) because it has been validated for streams under conditions similar to our case study, including comparable water discharges, grain sizes, and slopes.

Line 156: you start with the MPM equation, then present the Recking Eq. and then come back to MPM line 166. Rearrange this paragraph to make it shorter.

This paragraph has been reorganised (Line 172-190).

Line 203: not always the case in mountain streams, where the hydraulic radius R is preferred

The hydraulic radius (R) is typically preferred for streams with widths narrower than 20 times the river depth during sediment transport. However, this condition does not apply to the Gürbe River in the current situation, and we infer the same for non-engineered conditions. Additionally, we conducted tests using the hydraulic radius in our calculations, and the deviations from the simplified formulation are negligible, primarily due to the uncertainties associated with other parameters.

Line 222-224: ?? I do not really understand this sentence. It is the case for any equation if you consider computation for steady state condition.

This is the case for both equations applied in this work, therefore we removed this statement here. In Section 3.1, we assume uniform flow in our equations.

Line 241: you finally use 2 different equation for computing the hydraulics and the Shields number. Later in the paper you compare the results of the bedload equations. But the differences may come not from these equations but from large differences in the Shields for given discharge? The discussion should plot a comparison between the shield stresses obtained by these two methods.

The differences in the Shields number ($\tau^*$) for both equations mostly arise because each employs a different method to calculate the water depth (d). However, these differences are not significant, as water depth values do not vary substantially between the two methods. The primary distinctions between the formulations stem directly from the estimation of the dimensionless sediment bedload flux ($\Phi$) (Eq. 2 for MP.M. and Eq. 8 for Recking). Therefore, the main differences can be accurately assessed by examining the bedload flux itself, given that the transformation from $\Phi$ to the dimensional bedload flux is linear (Eq. 1).

Line 251: it was not possible to measure directly the grain size of the colluvial material (or of the transported material deposited at the outlet of reach 2)?

We were unable to measure the grain sizes of the colluvial material. The grains measured at site 13 actually constitute the material deposited at the outlet of reach 2. However, this site is consistently affected by human interference, and even there, fines are removed during flooding, making it unrepresentative of the colluvial material.

Lien 298: replace "the transport of the bedload" by "bedload transport"

Suggestion accepted.

Line 314: this paragraph is not clear for me. What is measured at the Burgistein station? If you have the flow chronicle, you can just convert with a Surface ratio (with power exponent 0.8) as often done. Why do you need precipitation? It is also the first time I see the scaling of Eq.12; do you have a reference?

At the Burgistein station, we have the flow chronicle. The power exponent for the surface ratio can vary widely across different catchments, and we lack a reliable reference for this parameter in our specific catchment. Consequently, we opted to use the precipitation ratio, considering it as a real constraint for our catchment. However, we can convert our precipitation ratio formulation into the power exponent, yielding values ranging between 0.9 and 0.95. We incorporated this discussion into the manuscript (Lines 353-362).

Line 339: could you explain the workflow used for calculating the uncertainty propagation? Or add a diagram in supplementary material...

The uncertainty propagation workflow comprises two main steps: uncertainties of the initial parameters and bedload fluxes. Each parameter (grain size, slope, and width) follows a specific workflow:

**Grain Sizes**:
- We employed the method developed by Mair et al. (2022) for grain sizes, utilizing survey-specific Structure-from-Motion (SfM) uncertainties (Table S1). This involved a routine combining bootstrap and Monte Carlo simulation, conducted separately for each river bar.
- In each bootstrap scenario (n = 1000), we resampled all measured grains, simulating distortions related to survey-specific SfM for each grain (details in Mair et al., 2022). The D50 and D84 were stored for each scenario.
- After the bootstrap, we fitted all stored D50s and D84s into a gamma distribution for a subsequent Monte Carlo simulation on bedload flux calculations. We justify the use of the gamma distribution due to the non-normality of the D50s and D84s distributions. The median, as well as the 95% confidence interval, were calculated for D50s and D84s.

**Slope**:

- Slope uncertainties were calculated by applying a bootstrap (n = 1000) on measurements of each surveyed site. For each scenario, the median slope was stored.
- After the bootstrap, we calculated the 95% confidence interval and fitted a normal distribution for the slope.

**Widths**:
- Width uncertainty was estimated by assuming a uniform distribution, with the range of the distribution defined by the ± 10% value of the measured width at each site.

**Bedload Flux Uncertainties**:

- Calculated using a Monte Carlo simulation (n = 1000), where each scenario calculated the bedload flux using input parameters sampled from random variables (rsv function in Python) following the distributions estimated in the previous step.
- Confidence intervals of the calculated bedload fluxes were determined from these scenarios.

We added this workflow in section S2 of the supplementary material.

Line 353: it seems logical if the stream channel is filled with materials

Line 373: Nice figure. D50 show a bimodal distribution. How do you explain the increase from 6 to 10? Lateral iputs?  Fig4e not commented in the text..

Upstream of the apex, lateral inputs induce changes in grain sizes. However, downstream of the apex, there are no more lateral inputs. We have two hypotheses for the observed coarsening in grain size on the alluvial fan: (1) It is not a real trend; the coarser reaches were recently washed from the fine material, while the fine-grained reaches are in a depositional state, showing the buffering effect of the check dams; (2) The other hypothesis is that this trend is related to variations in the channel width of the surveyed reaches (sites 8 to 11), where the widths are narrower than the sites 6-7 and 12-13 (Table S2). This condition makes the narrower reaches have more capacity for transport, reducing the probability of fine grains being deposited.

Fig 4e was initially planned to provide a reference for the location of the surveyed sites. We cited it in the text (Line 414).

Line 404-408: 50% lower in other sites? The figures shows near zero values

We rephrased this sentence (Line 433). The MP.M. equation predicts values of mean annual bedload fluxes below 2'500 m$^3$ y$^{-1}$ for all the other sites upstream of the apex.

Line 438-455: remove (repetition of lines 404-421)

Thank you for letting us know, we removed the repeat.

Line 458-466: this paragraph is not clear at all. Mainly because the results and method are in supplementary material. And it is not clear also what the objectives are. Obviously what is compared here is not the bedload equations but the flow resistance equations (see my comment above)?

We agree. This paragraph has been rewritten to make the objectives clearer (Lines 467-475).

In this paragraph, our aim is to highlight that, for engineered conditions, not all surveyed reaches can entrain the D84 grain size percentile during a strong peak discharge. Conversely, under non-engineered conditions, all currently engineered sites have the capability to mobilize the D84, ultimately disrupting the riverbed armouring.

We justify our decision to present these findings in the supplementary material as they represent a variation of the results depicted in Fig. 8, where the D50 grain size percentile served as the threshold in the Shields equation.

Line 485: maybe one way to verify the consistency could also compare the production m3/km² with what has been measured in other catchments? (see https://nhess.copernicus.org/articles/23/1769/2023/)

Thank you for providing this reference. We acknowledge that we haven't presented any comparisons for the non-engineered predictions in the Gürbe with other works. In the suggested paper, the dataset includes examples of catchments with areas and average channel slopes similar to our non-engineered condition. The mean annual sediment volumes for some of these catchments are of the same order of magnitude as our predictions using Recking equations but never align with the MP.M. predictions. This serves as compelling evidence that the Recking equation provides more reliable predictions for the non-engineered condition than the MP.M. equation. We incorporated this valuable information into section 5.1 (Lines 498-500).

Line 486-497: this part seems a justification for the use of these equations, which has already been done in the "methods" part

In this part, we justify why the predictions using the Recking equation are of better quality for the non-engineered situation than the MP.M. equation. We removed this paragraph.

Line 567: reformulate "Were it the case that the Guerbe riverbed was not stabilized"

"In a scenario where the Guerbe riverbed has not been stabilized, fluvial erosion could lead to an increase in sediment supply by activating shallow-seated landslides."

Line 578:this is true for both equations no?

No, the Recking equation predicts a continuous increase in the bedload flux along the sediment supply area (Fig. 7d). In contrast, the MP.M. equation predicts a strong increase in the bedload flux between sites 1 and 2 and keeps constant until the alluvial fan apex (Fig. 7b).

---

## Author Response (AR2)

I was satisfied by the authors answers to my comments and how the amended the text. I suggest some little additional modifications authors must feel free to consider or not.

We appreciate your insightful second review. Your constructive comments were valuable in enhancing our manuscript during the review process.

Line 18 : I suggets replacing 'Meyer-Peter Müller versus Recking approach' by 'Meyer-Peter Müller and Recking'

Suggestion accepted.

Line 20 : replace 'the firts equation is a classical' by 'the first equation is derived from a classical'

Suggestion accepted.

Line 21 : replace 'based on bedload data acquired from active mountain streams under steeper conditions' by 'based on a bedload field data set comprising active mountain streams under steeper conditions'

Suggestion accepted.

Line 138 : replace 'all available equations' by 'all equations'

Suggestion accepted.

Line 140 : suggestion replace: 'Despite this limitation, these equations were adjusted to represent the average bedload flux under the same boundary conditions, proving to be a powerful tool for estimating sediment flux over long-time scales. Therefore, the published and thus available bedload equations potentially serve as a suitable tool for our study, which focuses on relative changes in sediment transport capacity between engineered and non-engineered conditions in the Guerbe River.' ->'Despite this limitation, these equations were adjusted to represent the average bedload flux under various boundary conditions, and, therefore, remains relevant tools when used in a relative maner for estimating changes in sediment transport capacity between engineered and non-engineered conditions in the Guerbe River.'

Suggestion accepted.

Line 132 and 145 : remove 'two different equations representing the end members of a large panel of' line 132 and line 145 replace 'Among the various bedload equations that have been published in the scientific literature, two equations turn out to be most suitable for our basin. These are, as argued for below, the Meyer-Peter and Müller (1948) and Recking (2013) formula.' by 'Among the various bedload equations that have been published in the scientific literature, we chose to consider the Meyer-Peter and Müller (1948) and Recking (2013) equations, as they are representative for the two families of equations derived respectively from flume and field data»

Suggestion accepted.

Line 165 : 'occurrence of a near uniform flow'.

Suggestion accepted.

Line 174-178 : you already mentioned that. remove this sentence

Suggestion accepted.

Line 198 : remove 'better'

Suggestion accepted.

Line 210 : suggestion 'For both the MP.M. and Recking approaches we computed the dimensionless sediment bedload flux, with the Einstein parameter (Φ) (Einstein, 1950):'

Suggestion accepted.

Line 373 : what do you mean by 'and is equivalent to a power exponent,' ? It produces results similar as when applying a power exponent ?

Yes, our ratio approach yields comparable results to those obtained by applying a power exponent ranging between 0.9 to 0.95.

Line 414 replace 'data on grain sizes of sedimentary particles on the riverbed surface' by ''data on bed surface grain sizes'

Suggestion accepted.

Figure 6, 7, 8: In mountain streams, unlike alluvial rivers, transport depends on the availability of material which is not always the case. What you compute in more representative of a transport capacity. This explains why you compute greater transport at the appex than downstream, which questions on the continuity of transport along the profile. This does not change your results, but should be more clairly mentioned. Besides, Line 522 you write 'The application of two different approaches to calculate the bedload transport capacity'…

Thank you for your suggestion. We acknowledge that the assumption regarding material availability was not explicitly stated in the text. To address this, we inserted the following sentence in lines 195-197: "Furthermore, it is important to note that in mountain streams, the amount of transported material is influenced by its availability. In this study, we clarify that bedload flux calculations are interpreted as the transport capacity of the material under specified boundary conditions."

The scarcity of available material during a sediment transport event results in a change in the stream slope, leading to a condition of reduced transport capacity. In engineered conditions, the slope of the reaches between the dams exhibits significant fluctuations closely associated with material availability (e.g. between the apex in site 7 and downstream sites). This mechanism is detailed in Section 5.2, referred to as the regulation of sediment transport.

Line 667 after 'Both equations resulted in similar predictions regarding mean annual bedload fluxes for the currently engineered state. In contrast, models… ' Suggestion : add 'both models also predicts higher transport in the non engineered state. However, models…'

Suggestion accepted.